# The hidden cost of a smartphone: The effects of smartphone notifications on cognitive control from a behavioral and electrophysiological perspective

Joshua D. Upshaw[1]*, Carl E. Stevens, Jr.[1], Giorgio Ganis[2], Darya L. Zabelina[1]

1 Department of Psychological Science, University of Arkansas, Fayetteville, Arkansas, United States,
2 School of Psychology and Brain Research and Imaging Centre (BRIC), Plymouth University, Plymouth, Devon, England

* jupshaw@uark.edu

**Data Availability Statement:** We have created a publicly available OSF repository that includes the R scripts and data files to improve transparency.

## Abstract

Since their release in 2007, smartphones and their use have seemingly become a fundamental aspect of life in western society. Prior literature has suggested a link between mobile technology use and lower levels of cognitive control when people engage in a cognitively demanding task. This effect is more evident for people who report higher levels of smartphone use. The current study examined the effects of smartphones notifications on cognitive control and attention. Participants completed the Navon Letter paradigm which paired visual (frequent and rare target letters) and auditory (smartphone and control) stimuli. We found that overall, participants responded slower on trials paired with smartphone notification (vs. control) sounds. They also demonstrated larger overall N2 ERP and a larger N2 oddball effect on trials paired with smartphone (vs. control) sounds, suggesting that people generally exhibited greater levels of cognitive control on the smartphone trials. In addition, people with higher smartphone addiction proneness showed lower P2 ERP on trials with the smartphone (vs. control) sounds, suggesting lower attentional engagement. These results add to the debate on the effects of smartphones on cognition. Limitations and future directions are discussed.

## Introduction

"My favorite things in life don't cost any money. It's really clear that the most precious resource we all have is time."- Steve Jobs

Acknowledging the global prevalence of smartphone use requires little convincing, given the ubiquity of text messaging, selfie liking, and unlimited access to information at the touch of a button [1]. According to the Pew Research Center, 94% of adults in advanced economies own a smartphone or a similar device [2]. Since the release of the first iPhone, a large body of research has investigated the social and psychological impacts of mobile technology. Smartphones are undoubtedly beneficial in many ways, such as connecting with loved ones and

This can be accessed here: https://osf.io/bj7zf/ In addition, we created a publicly available github repository for the project where the data and other coding script materials will be made available upon acceptance of this manuscript for publication. https://github.com/jupshaw/SMARTPHONES-AND-COGNITIVE-CONTROL.

**Funding:** The authors received no specific funding for this work.

**Competing interests:** The authors have declared that no competing interests exist.

supporting our productivity goals [3]. However, smartphone use has also been demonstrated to have negative influences in a number of important life outcomes, such as "real-world" social interactions [4], walking and driving abilities [5, 6], and educational outcomes [7, 8]. Other research demonstrates a negative association between smartphone overuse (i.e., more than 2 hours per day) and psychological well-being [9]. For example, frequent social media use was found to be associated with a greater likelihood of developing severe anxiety [10], behavior and attention problems [11], and increasing the risk of suicide-related outcomes [12]. However, after accounting for other lifestyle factors (e.g., sleep, exercise, diet), the negative associations between smartphone use and well-being were rather small [13].

## Cognition and smartphone use

What may be common to all the aforementioned influences of smartphones described in the literature is their influences on people's executive functions, and namely their attention and cognitive control. From the framework of the cognitive load theory of attentional control [14], people with better cognitive control should be better at maintaining focus on the task at hand when exposed to task-irrelevant smartphone notifications. A recent review outlined multiple studies which have examined the relationships between mobile technology use and cognitive functions with the majority of studies indicating that increased smartphone use has been associated with reduced performance on tasks assessing cognitive control and attention [15]. In one study for example, heavy smartphone users were found to have a lower capacity for sustained attention during an arithmetic task [16]. Other studies found an increase in error rates on cognitive control tasks in people who use smartphones and social media more frequently [17–19]. Other work has demonstrated that people with higher media multitasking behavior (i.e., engaging in multiple forms of media concurrently) are worse at filtering out irrelevant distractor stimuli [20, 21], and show heightened attentional impulsivity [22, 23]. In terms of smartphone notifications, Stothart and colleagues [23] found that receiving a notification resulted in decreased sustained attention abilities similar to when people were actively using their devices. Beyond actually using smartphones and hearing notifications, prior studies have found that even the mere presence of smartphones can negatively affect performance on attention tasks [24, 25].

Although there is some behavioral evidence for the link between smartphone use and cognitive control, the neurophysiological mechanisms of this association are poorly understood. Thus, the present study aimed to examine the effects of smartphone notifications on behavioral and neural markers of top down executive functions, namely cognitive control, and attentional processes known to play a role in stimulus orienting and categorization. Recent findings from neuroimaging research provide a broad understanding of the structural and spatial neural activity of cognitive control processes associated with smartphone use. For example, cognitive control functions and smartphone use have been separately associated with activity in similar reward processing regions of dopaminergic neural pathways such as the ventromedial prefrontal cortex and dorsolateral prefrontal cortex [26, 27]. Another study found that heavy multimedia users had reduced gray matter volume in the anterior cingulate cortex, known for its involvement in higher order cognitive control processes [28]. Furthermore, higher media multitaskers are reported to recruit more neural resources from top-down control networks during a sustained attention task when they are in the presence of distractor stimuli [19]. Though these findings offer insight into potential neural mechanisms involved in smartphone use, they do not provide causal explanations, nor do they employ temporally specific measures necessary for understanding millisecond-level changes in cognitive processes as a function of smartphone use and exposure to notifications.

## Present study

The present study aimed to examine the extent to which smartphone notifications influence cognitive control and attention on an adapted Local/Global hierarchical letter three-stimulus oddball paradigm using event-related potentials (ERPs), and behavioral performance. The Local/Global task requires an individual to attentionally reorient to and update working memory to accurately respond to the presence of a target letter while monitoring for the presence of rare distractor letters presented at opposite levels of visual attention [29, 30].The rare target letter presents an exogenous salient singleton requiring increased recruitment of attentional and cognitive control resources, specifically those necessary for conflict monitoring [31]. Cognitive control paradigms such as the Stroop or Erickson Flanker task also measure conflict monitoring, however, the paradigm in the current study was chosen to heighten engagement of early attentional orienting mechanisms. While monitoring for conflict between frequent, rare, and non-target trials, participants were required to ignore inconsistent visual information between hierarchically nested visual stimuli presented at opposing levels of local or global attention. Therefore, this paradigm allowed us to simultaneously measure the effects of smartphone notifications on attention and cognitive control.

Cognitive control was measured using the *oddball effect*, which is calculated by subtracting reaction times (RT) and ERP amplitudes on frequent target trials from rare target trials. Better cognitive control is considered to be reflected by a smaller RT oddball effect and a larger ERP oddball effect [32]. We examined three ERPs, the P200 (P2), N200 (N2), and P300 (P3), which are commonly accepted in the literature as underlying neural markers of electrical cortical activation associated with attention and cognitive control processes [30, 33, 34].

This paradigm and various adaptations have been used in prior studies indicating a family of frontocentral N2 components related to cognitive control [33]. The N2 ERP component is the second negative peak along the average ERP waveform which generally occurs between 200–350 ms after stimulus onset near frontocentral and central electrode site. N2 is considered to be involved in strategy regulation, feedback processing, immediate action control, novel stimuli detection, and visual attention orienting [33]. Though multiple N2 subcomponents exist, the current study focused on a frontocentral N2 component related to cognitive control processes of response inhibition, response conflict, and error monitoring. This anterior N2 component is said to be generated from the anterior cingulate gyrus [35] and is associated with top-down control of attention [36]. Based on previous literature, we expected that participants would respond more slowly and would show a smaller N2 oddball effect (i.e., worse cognitive control) on trials with the smartphone notification (vs. control) sounds.

We examined the P2 ERP as it is likely to be affected by exposure to smartphone notifications. The P2 ERP is the second positive peak along the average ERP waveform which generally occurs between 150–250 ms after stimulus onset near frontal electrode sites [34]. P2 is considered to reflect stimulus monitoring and early attention classification processes and has been shown to demonstrate differential activation between target stimuli conditions in oddball paradigms assessing capacities to withdraw attentional resources away from stimuli [37, 38]. P2 is said to be generated largely as a result from activation within the reticular activating system [39] as a response to input from sensory modalities [34]. If it is the case that smartphone notifications "capture" people's attention, trials with smartphone notification (vs. control) sounds should elicit a larger P2 ERP.

Among oddball paradigms, an anterior N2 component to frequent targets is often observed in combination with a posterior P3 component to distractor targets, suggesting cognitive processes involved in contextual and memory updating [40, 41]. The P3 ERP component is the third positive peak along the average ERP waveform which generally occurs between 250–500 ms after stimulus onset near large frontoparietal scalp electrode networks [30]. P3 can be

divided into two subcomponents. P3 is said to originate from frontal lobe activation for attention-driven stimuli processing, particularly for task-irrelevant neural activity elicited during target stimulus processing [30]. P3 has been considered to be a late cognitive component involved in endogenous decision-making and stimulus categorization originating from the dorsolateral prefrontal cortex (DLPFC) which communicates with the cingulate cortex and parietal structures [35]. P2 and N2 reflect early cognitive processes and are likely to be influenced by exogenous smartphone notifications. P3, on the other hand, should not be affected by smartphone notifications, as P3 is thought to reflect late cognitive processes involved in endogenous decision-making and stimulus categorization [32].

We also considered the role of individual differences in self-reported smartphone addiction. According to Folk and colleagues [42], attentional orienting is dependent upon one's internal attentional control settings, which are dictated by people's current behavioral goals and are subject to being influenced by their individual cognitive biases. Thus, people who use smartphones more frequently, should be more likely to have an underlying cognitive bias to orient their attention to their devices when hearing smartphone notification sounds. In contrast, people who use their phones less frequently should demonstrate a greater ability to complete behavioral goals as they are less cognitively biased to orient to their devices when hearing notifications.

In fact, studies have shown that people with excessive (vs. moderate) smartphone use show higher N2 during a Go-NoGo task in which participants were asked to view smartphone-relevant (vs. control) images [43]. The authors suggested that excessive (vs. moderate) smartphone users recruited higher levels of cognitive control necessary for inhibition processes in order to maintain similar levels of goal-directed behavioral performance when exposed to smartphone related stimuli. Additional work found that people higher (vs. lower) in problematic smartphone use, measured by the Smartphone Addiction Proneness Scale, showed smaller N2 amplitudes, delayed response latencies, and higher error rates on a Go-NoGo task [44]. This effect was found to be stronger when participants were exposed to smartphone notification vibration sounds. Thus, based on previous studies, we expected that individual differences in proneness to smartphone addiction would be associated with slower responses and worse cognitive control (larger RT and smaller N2 oddball effect) on trials with smartphone (vs. control) sounds.

## Methods

### Ethics statement

The study was approved by the local Institutional Review Board at the University of Arkansas and was assigned the protocol number 1807134340. All participants provided written informed consent to participate. Participants were compensated with course credit.

### Participants

College students ($N$ = 73; 38 male, 35 females, mean age = 19.78, $SD$ = 0.32, 80% white) participated for course credit. Participants were daily smartphone users, had normal or corrected to normal hearing and vision, and were fluent English speakers. Participants had no history of brain damage or concussions and were not currently diagnosed with any psychiatric disorders. Participants were instructed to not be under the influence of excessive caffeine, unprescribed medication, or alcohol at the time of testing.

### Materials

**Oddball paradigm.** A Local-Global Navon letter task [45] was used to measure the oddball effect. This task is identical to the one employed by earlier work [32]. The letter stimuli

were designed to elicit approximately equal response speed and accuracy for both global and local letters [46]. In addition, all letter stimuli possessed similar perceptual features [47] reducing concern for stimulus orienting contingency on attentional control settings [42]. Further, this paradigm was designed such that motor-related confounds for ERPs are reduced as both target and non-target responses require similar motor activity [48]. The visual stimuli consisted of twelve composite letters of one global letter comprised of several uniform (never mixed) local letters (Fig 1A). The local letters (subtending 0.43 by 0.86 degrees of visual angle) were arranged within a 5 cm x 3 cm rectangular grid to form the global letters (subtending approximately 2.57 by 4.27 degrees of visual angle).

Participants were asked to indicate the presence of a target letter by pressing either Yes (1 key) for "Target letter is present," or No (2 key) for "Target letter is not present" using their right hand on a standard keyboard digit pad. Participants were instructed to detect the presence of lack of presence of the target letter regardless of the size of the letter. Participants completed two practice blocks consisting of 9 trials each. Visual feedback was provided for response accuracy on practice trials ("Correct" or "Incorrect"). Following practice trials, participants completed 16 experimental blocks (960 trials). Before each block, a "target letter" screen was displayed on the computer monitor for 12 seconds. The "target letter" screen used a single red letter (twice in size as the local letters) to identify the specific target letter participants would be aiming to detect in the following block of trials.

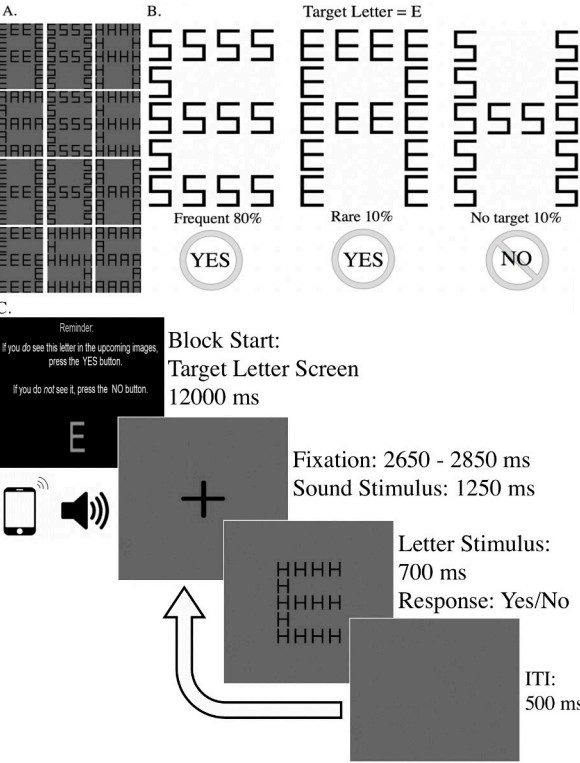

**Fig 1. Modified Navon letter oddball paradigm.** A): Twelve composite letters: a global *A* made of local *E*'s, *S*'s, or *H*'s; a global *E* made of local *A*'s, *S*'s, or *H*'s; a global *H* made of local *E*'s, *S*'s, or *A*'s; and a global *S* made of local *E*'s, *H*'s, or *A*'s. B): In this example (white background used for example), participants were instructed to determine if the target letter E is present (either at the global or local level, 80% and 10% of trials, respectively), or not present (10% of trials). C): Single trial structure of the oddball task. ITI = inter-trial interval.

Each block of the task consisted of 60 trials presented in pseudo-random order, ensuring that an equal number of sound stimuli were presented on frequent, rare, and non-target trials. On a given block, target letters were displayed on the screen at either the local or global attentional level on 80% of trials, referred to as frequent trials. During the same task block, target letters of the opposite level of attention were displayed on 10% of trials, referred to as rare trials. The final 10% of trials did not include a target letter, referred to as non-target trials; (Fig 1B). Global and local letter stimuli were counterbalanced such that big "H"s, "E"s, "S"s, and "A"s, were composed of uniform (never mixed within a single letter) sets of smaller letters for an equal presentation of hierarchical letter combinations across the 4 four possible letters to choose from. For example, on a given trial, a big "H" would be comprised of all small "S"s, but never small "S"s and small "E"s. Each block of 60 individual trials was followed by a self-paced break period.

Each composite letter stimulus was preceded by one of three auditory stimuli: A smartphone notification sound, a lawnmower sound, and a computer-generated control sound. The smartphone notification sound was the sound of the vibration of smartphone when it receives an incoming notification. In a separate pilot study, 97% of participants correctly identified the sound as the smartphone vibration. For the control sound, 14% indicated it was a smartphone sound, 26% said it was a food processor, and 60% said they could not identify the sound. Furthermore, 63% of participants indicated that their smartphone's notification setting was typically set to vibrate (3% on sound, 33% on silent, 1% other), and 70% reported their text notification setting was set to vibrate most of the time (1% on sound, 29% on silent). Finally, 1% of participants said the smartphone notification was exactly the same as their own, 30% said it was nearly the same, 55% said it was somewhat similar, and 14% of participants said it was not at all similar.

The control sound was created in Audacity (v. 2.2.2, [49]), and was a square wave tone closely matched to the smartphone sound in duration, volume, and sound similarity. The sounds were delivered via noise canceling insert earphones at 70 percent maximum volume within safe listening levels (~ 60 dB). The lawnmower sound was included as an additional control stimulus for sound familiarity (i.e., smartphone vs. ambiguous control sound compared to smartphone vs. lawnmower sound). Consequent examination of the lawnmower sound acoustic waveform spectrum revealed unintended technical confounds (e.g. stereoscopic inconsistency creating a perception of spatial movement (Fig 2C) and was not used in subsequent analyses. Sound stimuli were presented pseudo-randomly.

There were 960 trials in total: 768 frequent target trials (256 per sound condition), 96 rare target trials (32 per sound condition), and 96 no target trials (32 per sound condition). On each trial, a sound was played (~1250 ms) concurrently with a randomly jittered fixation cross (2650–2850 ms), followed by the letter stimulus (700 ms), during which time participants indicated the presence or absence of the target letter. If participants failed to respond within the 700 ms period, the task timed out and moved on to the next trial. A uniform gray screen appeared for 500 ms during the inter-trial interval before the next trial began (Fig 1C).

Four participants were excluded from behavioral data analyses due to technical issues or poor performance on the oddball task (i.e., errors or RTs exceeding +/- 2.5 SD). The final sample for behavioral analyses included 69 participants. ERP data for 19 participants were excluded because of technical issues, or for having uncorrectable artifacts greater than 25% of total trials [48]. The final sample for ERP analyses included 54 participants.

**Smartphone addiction.** The Smartphone Addiction Proneness Scale (SAPS) [50], is a 15-item self-report questionnaire that assesses individual differences in smartphone addiction proneness. The scale consists of four factors: Disturbance of adaptive functions (e.g., "My school grades dropped due to excessive smartphone use."), virtual life orientation (e.g., "When

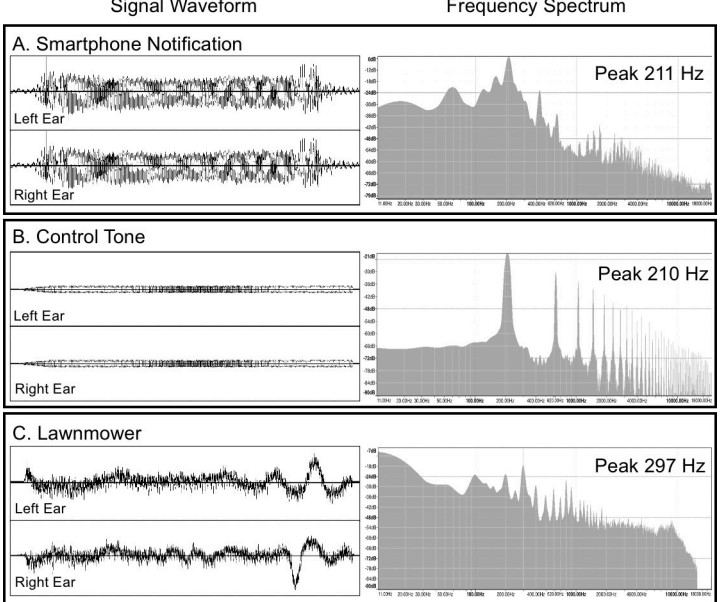

**Fig 2. Frequency spectral densities and signal waveforms of the auditory stimuli.**

I cannot use a smartphone, I feel like I have lost the entire world."), withdrawal (e.g., "It would be painful if I am not allowed to use a smartphone."), and tolerance (e.g., "I try cutting my smartphone usage time, but I fail."). Responses are made on a 4-point Likert-type scale and range from 1 (*strongly disagree*) to 4 (*strongly agree*). The total SAPS score is an average of the four factors, with higher scores reflecting higher levels of proneness to smartphone addiction. The SAPS was reported to have high construct validity (NFI = .943, TLI = .902, CFI = .902, RMSEA = .034) and reliability ($\alpha$ = .88).

**Electrophysiological recordings.** Continuous EEG data were sampled at 2048 Hz using a Biosemi Active Two system. EEG data were collected from 32 active Ag/AgCl electrodes arranged according to the 10–20 system. Two loose lead electrodes below both eyes monitored eye blinks, 2 on the outer canthi of the eyes monitored horizontal eye movements, and 2 on the mastoids were used as reference. EEG data were preprocessed in MATLAB (2018b) using the EEGLAB (v13.6.5b) toolbox before further processing [51]. Continuous EEG data were down-sampled off-line to 512 Hz and high pass Basic FIR filtered at .1 Hz.

**ERP waveform and component analysis.** EEGLAB and ERPLAB were used to process EEG data offline. ERPs were averaged off-line for a 1000 ms total epoch segment (200 ms pre-stimulus and 800 ms post-stimulus). Artifact detection was performed to assess trials contaminated with eye blinks, horizontal eye movement, muscle activity, or other signal noise. First, a moving window peak to peak artifact detection threshold was used on vertical eye channels (voltage threshold = 75 μV, moving window width = 200 ms). Second, a step-like artifact detection analysis on lateral eye channels (voltage threshold = 30 μV, moving window width = 400 ms, window step = 10 ms). Third, a moving window peak-to-peak threshold was used on all the channels (voltage threshold = 200 μV, moving window width = 1000 ms). ERPs were 2nd order (IIR) Butterworth low-pass filtered at 30 Hz (12 dB/octave roll-off).

Participants with greater than 25% overall artifact rejections were reprocessed using independent component analysis (ICA) and bad components were inspected using ICLabel, an EEGLab plugin [52]. Non-brain identified components with greater than 94.5% confidence were removed from the data. ICA-corrected data were re-processed and artifact detections

were repeated. After reprocessing ICA-corrected participants, people with greater than 25% overall rejection thresholds were excluded from data analysis. Mean ERP amplitudes of ICA and non-ICA-corrected participants did not differ, independent samples $t$-tests, $p$s > .064.

Past literature provides reasonable consensus for the ERP latencies implicated in cognitive control. However, the defined time windows can and arguably should vary across studies depending on variations in study design, stimuli, tasks, participants, conditions and unknown noise [53, 54]. Thus, a data driven approach to defining ERP latency windows provides an optimal solution for accounting for these variations in temporal and spatial location of condition effects, reducing Type I and II error rates. Precise parameters for the ERP component temporal latency windows were determined through the grand average of conditions across participants. Peak latency analyses were performed for each component at their maximal channel sites. P2, N2, and P3 mean amplitudes were measured at channel sites Fz (150–210 ms), F3 (220–380 ms), and Pz (260-700ms), respectively [34, 33, 30]. Latency windows based on prior literature were applied to detect peak amplitudes and then visually inspected to capture the entire ERP component. To avoid component overlap, a 10 ms time window was used to separate component latencies [55]. A mean amplitude value between these latencies was calculated for subsequent analyses.

The oddball effect was computed for each ERP component by subtracting the mean amplitude on frequent target trials from rare target trials. P2 is a positive-going component, thus *larger positive* values indicate a larger oddball effect, reflecting better stimulus monitoring and early classification processes. The N2 is a negative-going component, thus *smaller positive* values indicate a larger oddball effect, reflecting better cognitive control. P3 is a positive-going component, thus *larger positive* values indicate a larger oddball effect, reflecting better task relevant stimulus categorization processing.

## Procedure

Eligible participants volunteered to participate in the study. Participants were positioned 67cm from the center of the computer screen and received instructions for the oddball task. They were instructed to respond quickly and accurately on all trials. To reduce EEG artifacts, participants were asked to minimize blinking, facial, and bodily movement throughout the task. After two practice blocks (18 trials), participants completed 16 blocks of the oddball task while EEG data were recorded continuously. Each block lasted approximately 5 minutes, with a 75-minute approximate total task time. After the task, participants completed questionnaires via Qualtrics. The entire session lasted approximately 150 minutes.

## Analytical strategy

Data were analyzed in RStudio [56]. For RTs, analyses were conducted using linear mixed effect regression (LMER) models with random slopes for the trial frequency condition (rare and frequent) and random intercepts for each participant to account for within-subject variance in RT across all trials (lmerTest v. 3.1.1) [57]. This model was found to provide the best fit of the data relative to simpler LMER models. An intraclass correlation (ICC) of 0.14 for RT within participant was found, warranting the use of mixed linear models. We assessed differences in RT on rare vs. frequent target trials to evaluate the presence of oddball effect, then determined if RT varied as a function of trials with smartphone notifications vs. control sounds. Then, we assessed if differences in the oddball effect were present as a function of the sound condition. A three-way interaction regression was then performed to assess the moderating role of smartphone addiction in the effects of smartphone notifications on cognitive control.

For the accuracy analyses, we conducted generalized linear mixed effects regression model with error rate as the dependent variable and random slopes for trial frequency and random intercepts for participant. We used a bound optimization by quadratic approximation with a binomial family distribution of 0 for correct and 1 for incorrect trials.

For ERPs, paired sample *t*-tests were conducted to compare the grand averaged mean amplitudes of the P2, N2, and P3 components on rare and frequent targets (oddball effect), and on trials with smartphone and control sounds (trial to trial data were not available). Linear models were carried out to determine whether smartphone addiction scores predicted overall ERPs, overall oddball ERPs, and oddball ERPs between sound conditions (smartphone vs. control sounds).

## Results

### Behavioral findings

**Reaction time.**   Mixed model regression analyses revealed that, across all trials (N = 37,769), participants responded slower on rare (*M* = 476.99 ms, *SD* = 115.57) than on frequent trials (*M* = 426.83 ms, *SD* = 87.53), Cohen's *d* = 0.49, demonstrating a reliable 50.96 ms oddball effect with a medium effect size (Table 1). As predicted, participants responded significantly slower on trials with smartphone sounds (*M* = 432.97 ms, *SD* = 92.10) than on trials with control sounds (*M* = 429.70 ms, *SD* = 90.95), Cohen's *d* = .04, demonstrating a small effect size. There was no significant interaction between trial frequency and sound condition, indicating that cognitive control did not differ as a function of the sound condition.

Additional regression models were conducted separately for frequent (N = 34,372) and rare (N = 3,397) target letter trials, to determine the extent to which, if any, trial frequency played a role in the effects of the sound conditions on RTs. On frequent trials participants responded significantly slower on trials with smartphone sounds (*M* = 428.51, *SD* = 87.95) than control sounds (*M* = 425.15, *SD* = 87.08), Cohen's *d* = .04 (Table 2). On rare trials, however, participants did not differ in their speed on smartphone (*M* = 477.53, *SD* = 117.29) and control (*M* = 476.46, *SD* = 113.80) sound trials, Cohen's *d* = .01. These findings suggest that participants slowed down on smartphone trials on frequent, but not on rare trials.

**Individual differences in proneness to smartphone addiction.**   We then sought to determine the role of individual differences in smartphone addiction in response speed to smartphone notifications. We conducted a mixed linear model with RT as the outcome variable, trial frequency (frequent vs. rare trials), sound condition (smartphone vs. control sounds), and smartphone addiction (continuous, mean centered) as predictor variables. Results revealed that individual differences in smartphone addiction did not predict overall response speed (Table 3). There was no significant interaction between trial frequency and smartphone

**Table 1. Regression with reaction times predicted by trial frequency and sound condition.**

| Predictor | *b* | *SE* | 95% CI Lower | 95% CI Upper | df | *t* | *p* |
|---|---|---|---|---|---|---|---|
| Trial Frequency [a] | 50.96 | 3.74 | 43.53 | 58.39 | 68 | 13.63 | < .001 |
| Sound [b] | 3.18 | 1.51 | 0.23 | 6.13 | 37670 | 2.11 | .035 |
| Trial Frequency x Sound | -0.73 | 3.01 | -5.17 | 6.63 | 37700 | -0.24 | .808 |

Reaction times on the oddball task as a function of trial frequency (rare vs. frequent) and sound condition (smartphone vs. control).

[a] Trial frequency was contrast coded at -.5 for frequent and .5 for rare target trials.

[b] Sound condition was contrast coded at -.5 for control sound and .5 for smartphone sound. Dependent variable = Reaction time in ms. CI = Confidence Interval.

**Table 2. Regressions for reaction times for frequent and rare trials predicted by sound condition.**

| Predictor | b | SE | 95% CI Lower | 95% CI Upper | df | t | p |
|---|---|---|---|---|---|---|---|
| **Frequent Trials** | | | | | | | |
| Sound [b] | 3.54 | 0.90 | 1.79 | 5.31 | 37632 | 3.95 | < .001 |
| **Rare Trials** | | | | | | | |
| Sound [b] | 2.81 | 2.87 | -2.82 | 8.45 | 37699 | 0.98 | .327 |

Dependent variable: RT (ms). Reaction times on the oddball task for frequent and rare target trials separately as a function of sound condition (smartphone vs. control).

[b] Sound condition was contrast coded at -.5 for control sound and .5 for smartphone sound. Dependent variable = Reaction time in ms. CI = Confidence Interval.

addiction, revealing that the RT oddball effect (i.e. cognitive control) was not significantly different as a function of smartphone addiction levels. RT did not significantly differ between smartphone and control sound conditions as a function of individual differences in smartphone addiction. Finally, the three-way interaction between trial frequency, sound condition, and smartphone addiction was not significant.

Additional regression analyses were conducted separately for frequent and rare trials. On frequent trials, there was a significant two-way interaction between sound condition (smartphone vs. control) and smartphone addiction (Table 4 and Fig 3). Specifically, people with higher levels of smartphone addiction were significantly slower to respond on frequent trials with smartphone (vs. control) sounds. On rare trials, the two-way interaction between sound condition (smartphone vs. control) and smartphone addiction was not significant, indicating that response speed did not differ between smartphone (vs. control) sounds as a function of smartphone addiction levels on rare trials.

**Accuracy.** A generalized linear regression revealed that people made significantly more errors on rare ($N = 4065$, $M = 16.43\%$, $SD = 0.37$) compared to frequent trials ($N = 34806$, $M = 1.25\%$, $SD = 0.11$), while there were no significant differences in the number of errors made on trials with smartphone notifications ($N = 19442$, $M = 2.72\%$, $SD = 0.16$) compared to control sounds trials ($N = 19429$, $M = 2.95\%$, $SD = 0.17$; Table 5). These results indicate that accuracy on rare trials was worse than on frequent trials, as expected, however the presence of smartphone notification did not impair accuracy relative to control sounds.

**Table 3. Regression for reaction time predicted by trial frequency, sound condition, and smartphone addiction.**

| Predictor | b | SE | 95% CI Lower | 95% CI Upper | df | t | p |
|---|---|---|---|---|---|---|---|
| Trial Frequency [a] | 50.97 | 3.74 | 43.54 | 58.41 | 69 | 13.64 | .001 |
| Sound [b] | 3.19 | 1.51 | 0.24 | 6.13 | 37700 | 2.12 | .034 |
| SAPS [c] | 6.89 | 14.12 | -21.18 | 34.96 | 69 | 0.49 | .627 |
| Trial Frequency x Sound | -0.72 | 3.01 | -6.62 | 5.28 | 37700 | -0.24 | .811 |
| Trial Frequency x SAPS | -2.41 | 10.90 | -24.09 | 19.24 | 69 | -0.22 | .826 |
| Sound x SAPS | 1.94 | 4.38 | -6.64 | 10.52 | 37696 | 0.44 | .658 |
| Trial Frequency x Sound x SAPS | -8.53 | 8.76 | -25.69 | 8.63 | 37696 | -0.97 | .330 |

Reaction times on the oddball task as a function of trial frequency (rare vs. frequent), sound condition (smartphone vs. control), and smartphone addiction levels.

[a] Trial frequency was contrast coded at -.5 for frequent and .5 for rare target trials.

[b] Sound condition was contrast coded at -.5 for control sound and .5 for smartphone sound.

[c] SAPS = Smartphone Addiction Proneness Scale (mean centered). Dependent variable = Reaction time in ms. CI = Confidence Interval.

**Table 4. Regressions for reaction times on frequent and rare trials predicted by sound condition and smartphone addiction.**

| Predictor | *b* | *SE* | 95% CI | | df | *t* | *p* |
|---|---|---|---|---|---|---|---|
| | | | Lower | Upper | | | |
| **Frequent Trials** | | | | | | | |
| Sound[a] | 3.54 | 0.90 | 1.79 | 5.31 | 37631 | 3.95 | .001 |
| SAPS[b] | 8.10 | 11.74 | -15.24 | 31.43 | 69 | 0.49 | .493 |
| Sound x SAPS | 6.20 | 2.62 | 1.07 | 11.34 | 37632 | -0.24 | .018 |
| **Rare Trials** | | | | | | | |
| Sound[a] | 2.83 | 2.87 | -2.80 | 8.46 | 37700 | 0.98 | .325 |
| SAPS[b] | 5.69 | 17.90 | -29.89 | 41.25 | 69 | 0.32 | .752 |
| Sound x SAPS | -2.32 | 8.34 | -18.70 | 14.05 | 37698 | -0.28 | .781 |

Reaction times on the oddball task as a function of sound condition (smartphone vs. control) and smartphone addiction with separate results for frequent and rare trials.

[a] Sound condition was contrast coded at -.5 for control sound and .5 for smartphone sound.

[b] SAPS = Smartphone Addiction Proneness Scale (mean centered). Dependent variable = Reaction time in ms. CI = Confidence Interval.

Individual differences in smartphone addiction levels had no significant effect on error rates overall, *p* = .688, or in terms of responses between frequent and rare trials, *p* = .534, sound conditions, *p* = .793, nor their interaction, *p* = .391, showing that individual differences in smartphone addiction did not affect task performance in terms of accuracy.

## Neural findings

**ERP oddball effect.**　Paired sample *t*-tests were conducted to examine overall ERP differences between rare and frequent trials (Fig 4). P3 amplitude was significantly larger on rare versus frequent trials, Cohen's *d* = .29 (small to medium effect size). There were no significant differences in P2 or N2 amplitudes between rare and frequent trials (Table 6).

**ERPs between sound conditions.**　Paired sample *t*-tests were conducted to examine overall ERP differences between trials with smartphone and control sounds. P2 was

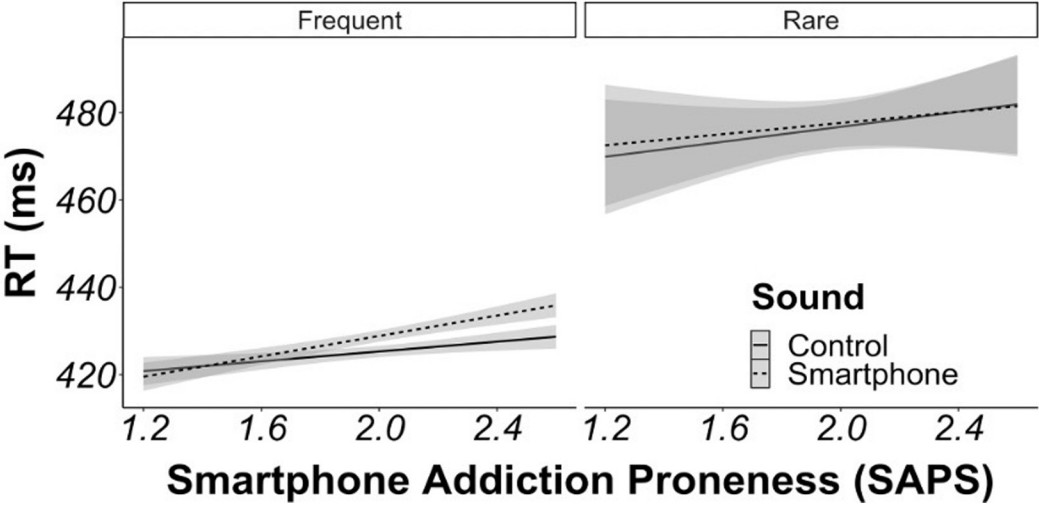

**Fig 3. Interaction analysis for reaction times predicted by sound condition and smartphone addiction.** Plot of reaction times on trials with smartphone sounds (dashed line) vs. control sounds (solid line) as a function of individual differences in the proneness to smartphone addiction. Plotted separately for frequent and rare target trials.

**Table 5. Logistic regression of error rates predicted by trial frequency and sound condition.**

| Predictor | Odds Ratio | $p>z$ | 95% CI | |
|---|---|---|---|---|
| | | | Lower | Upper |
| Trial Frequency [a] | 17.83 | 27.41*** | 2.68 | 3.10 |
| Sound [b] | 0.92 | -1.38 | -0.22 | 0.04 |
| Trial Frequency x Sound | 0.84 | .-1.36 | -0.43 | 0.08 |

Binomial logistic regression for the likelihood of an incorrect response.

[a] Trial frequency was contrast coded at -.5 for frequent and .5 for rare target trials.

[b] Sound condition was contrast coded at -.5 for control sound and .5 for smartphone sound. Dependent variable = Count of incorrect trials, (0 = Correct, 1 = Incorrect).

CI = confidence interval.

*** $p < .001$.

marginally smaller on smartphone versus control sound trials, Cohen's $d$ = .12 (Table 7). N2 was significantly larger on smartphone versus control sound trials, Cohen's $d$ = .13 (small effect size; Fig 5), suggesting that people generally had lower levels of cognitive control on trials with smartphone sounds. P3 did not significantly differ on smartphone versus control sound trials. These results indicate that overall, people had greater neural activation implicated in cognitive control on trials with smartphone notifications (vs. control) sound trials, while early and late attentional processes linked to stimulus orienting and categorization were not involved.

**ERP oddball effect between sound conditions.** Paired sample $t$-tests were conducted to examine differences in the ERP oddball effect (ERPs on rare minus frequent trials) between trials with smartphone and control sounds. The N2 oddball effect was significantly larger on smartphone versus control sound trials, Cohen's $d$ = .28 (small to medium effect size; Fig 6), pointing to the increased recruitment of cognitive control processes on smartphone trials. There was no significant difference in P2 or P3 oddball effect between the sound conditions (Table 8).

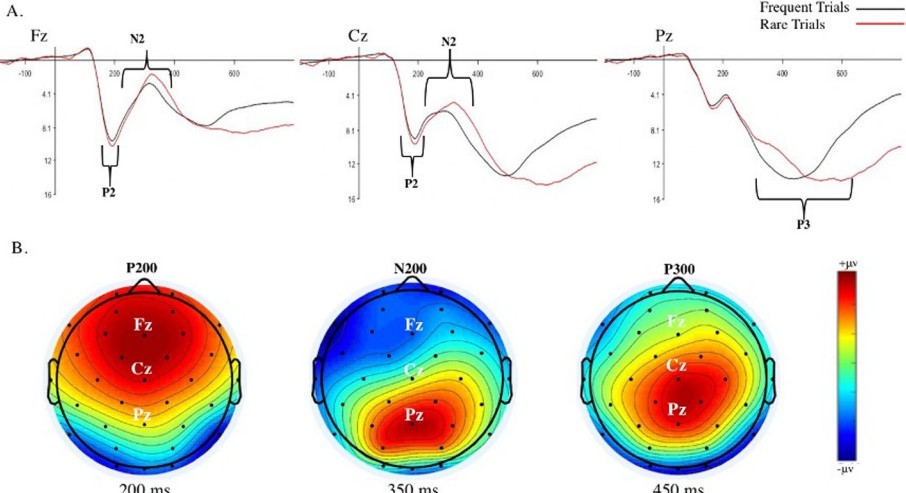

**Fig 4. Overall ERP waveforms and scalp maps.** A. Aggregated ERP waveforms for frequent (black lines) and rare trials (red lines). B. Aggregate ERP scalp distribution maps for P2 at 200 ms, N2 at 350 ms, and P3 at 450 ms latencies. Red color reflects activity for P2 and P3. Blue color reflects activity for N2. (Darker colors reflect increased activity).

**Table 6. Paired samples *t*-tests for ERP amplitudes between trial frequency.**

|  | *M* (*SD*) |  |  | 95% CI |  |  |  |  |
|---|---|---|---|---|---|---|---|---|
| ERP | Frequent Trials | Rare Trials | *Mdiff* | Lower | Upper | df | *t* | *p* |
| P2 at Fz | 8.45 μv (3.66) | 8.88 μv (3.98) | 0.42 | -0.06 | 0.91 | 53 | 1.74 | .088 |
| N2 at F3[a] | 4.24 μv (3.32) | 4.08 μv (3.94) | -0.16 | -0.72 | 0.40 | 53 | -0.58 | .567 |
| P3 at Pz | 10.66 μv (3.96) | 12.15 μv (6.10) | 1.50 | 0.58 | 2.42 | 53 | 3.26 | .002 |

Grand averaged ERP amplitudes (in microvolts) on the oddball task for frequent and rare trials. ERPs were time-locked to the presentation of the visual stimulus.

[a]N2 is a negative going potential, thus smaller values indicate larger ERP amplitude. Dependent Variable = ERP mean amplitude. CI = confidence interval.

**Overall ERPs and smartphone addiction.** Linear models examining overall ERP mean amplitudes with smartphone addiction as a continuous predictor variable revealed that people higher in smartphone addiction had a significantly smaller overall P2, $f^2$ = .09 (small effect size; Table 9 and Fig 7). N2 and P3 did not significantly differ as a function of individual differences in smartphone addiction.

**ERP oddball effect and smartphone addiction.** Linear models revealed no significant differences in P2, N2, or P3 oddball effects as a function of individual differences in smartphone addiction (Table 10).

**Overall ERPs by sound condition and smartphone addiction.** Linear models examining differences in mean ERP amplitudes as a function of trials with smartphone versus control sounds and smartphone addiction revealed no significant interaction between smartphone addiction and sound trials for P2, $b$ = 1.01, $SE$ = 0.66, $t(54)$ = -1.52, $p$ = .133, 95% CI [-2.33, 0.31], N2, $b$ = 0.23, $SE$ = 0.63, $t(54)$ = 0.37, $p$ = .715, 95% CI [-1.03, 1.49], or P3, $b$ = 0.16, $SE$ = 0.81, $t(54)$ = 0.20, $p$ = .841, 95% CI [-1.45, 1.77].

**ERP oddball effects by sound condition and smartphone addiction.** A linear model with the mean ERP amplitude oddball effect as a dependent variable and trials with smartphone versus control sounds and smartphone addiction as a continuous predictor variable revealed no significant interactions for P2, $b$ = 2.31, $SE$ = 1.47, $t(54)$ = 1.57, $p$ = .119, 95% CI [-0.59, 5.20], N2, $b$ = 0.54, $SE$ = 1.46, $t(54)$ = 0.37, $p$ = .710, 95% CI [-2.37, 3.48], or P3, $b$ = 2.09, $SE$ = 1.73, $t(54)$ = 1.21, $p$ = .232, 95% CI [-1.36, 5.54].

## Discussion

The current study investigated the effects of smartphone notifications on behavioral and neural markers of top down executive functions, namely cognitive control and attentional processes known to play a role in stimulus orienting and categorization. The study further aimed to examine whether these effects varied as a function of individual differences in self-reported proneness to smartphone addiction.

**Table 7. ERP amplitudes between the smartphone and control sound trials.**

|  | *M* (*SD*) |  |  | 95% CI |  |  |  |  |
|---|---|---|---|---|---|---|---|---|
| ERP | Smartphone | Control Sound | *Mdiff* | Lower | Upper | df | *t* | *p* |
| P2 at Fz | 8.44 μv (3.80) | 8.89 μv (3.83) | -0.46 | -0.92 | 0.01 | 53 | -1.98 | .053 |
| N2 at F3[a] | 3.94 μv (3.38) | 4.39 μv (3.78) | -0.45 | -0.89 | -0.02 | 53 | -2.09 | .041 |
| P3 at Pz | 11.42 μv (4.84) | 11.39 μv (5.08) | 0.03 | -0.53 | 0.58 | 53 | 0.09 | .927 |

Grand averaged ERP amplitudes (in microvolts) on the oddball task for frequent and rare trials. ERPs were time-locked to the presentation of the visual stimulus.

[a]N2 is a negative going potential, thus smaller values indicate larger ERP amplitude. Dependent Variable = ERP mean amplitude. CI = confidence interval.

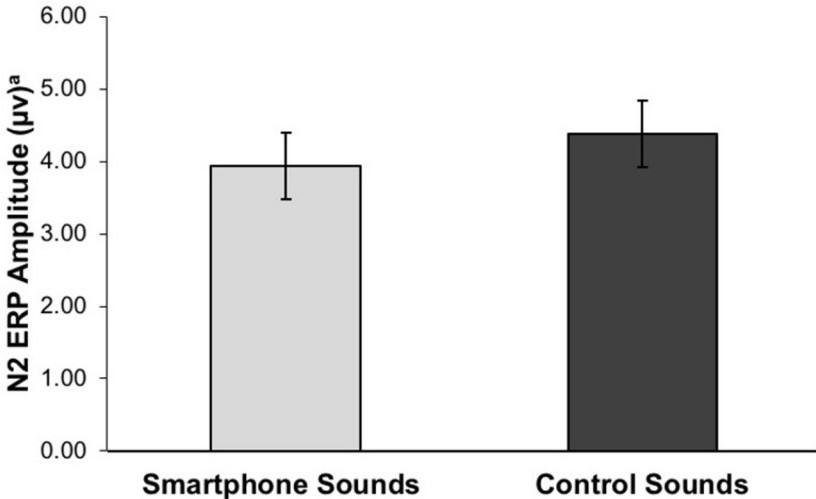

**Fig 5. N2 amplitude on smartphone and control sound trials.** N2 ERP mean amplitudes on trials with smartphone and control sounds. [a]N2 is a negative going potential, thus smaller values indicate larger ERP amplitudes.

## General effects of smartphone notifications on cognitive control

In line with our predictions, people responded slower on trials paired with smartphone (vs. control) sound notifications, although this effect was small. This finding is in line with recent work demonstrating, across two studies, that reaction times on trials with (vs. without) phone notifications were significantly slower [58].

These findings also offer partially contrasting evidence to previous literature which found that exposure to smartphone notifications on a sustained attention task [23] and Go/No-Go task [44] increased the speed of responding to target stimuli, which was also linked to an

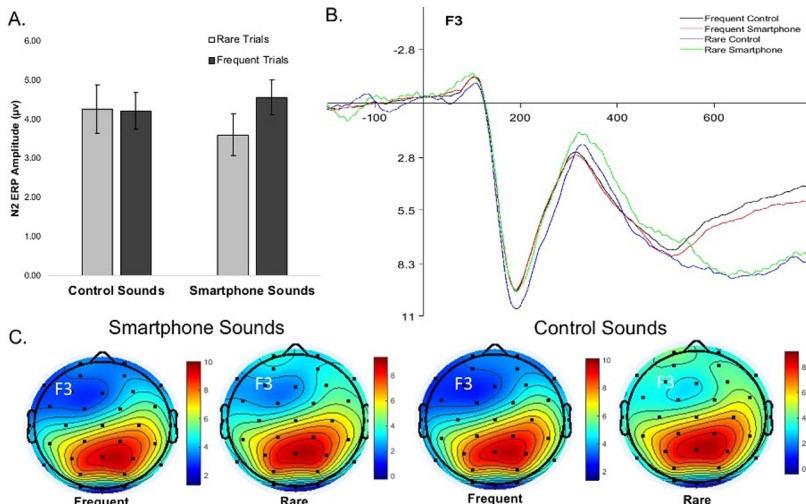

**Fig 6. N2 ERPs for trial frequency and sound conditions.** N2 ERP amplitudes on rare and frequent trials with smartphone and control sounds. The oddball effect is difference between rare and frequent trials. [a]N2 is a negative going potential, thus smaller values indicate larger ERP amplitude. A) Bar chart of mean N2 amplitudes. B) ERP waveforms for frequent trials with control sounds (black line), frequent trials with smartphone sounds (red line), rare trials with control sounds (blue line), and rare trials with smartphone sounds (green line). C) Scalp maps of N2 at 300 ms for frequent and rare trials after delivery of the sound stimulus.

**Table 8. ERP oddball effect between the smartphone and control sound trials.**

| ERP | M (SD) | | Mdiff | 95% CI | | df | t | p |
|---|---|---|---|---|---|---|---|---|
| | Smartphone Oddball Effect | Control Sound Oddball Effect | | Lower | Upper | | | |
| P2 at Fz | 0.09 μv (2.83) | 0.76 μv (2.48) | -0.67 | -1.75 | 0.40 | 53 | -1.26 | .213 |
| N2 at F3[a] | -0.67 μv (2.62) | 0.35 μv (2.88) | -1.02 | -2.03 | -0.01 | 53 | -2.03 | .047 |
| P3 at Pz | 1.47 μv (3.46) | 1.52 μv (4.53) | -0.05 | 1.16 | 0.58 | 53 | -0.08 | .936 |

ERP oddball effect amplitudes (in microvolts) on the oddball task for trials with smartphone and control sounds. ERPs were time-locked to the presentation of the visual stimulus.

[a]N2 is a negative going potential, thus smaller values indicate larger ERP amplitude. Dependent Variable = ERP mean amplitude oddball effect. CI = confidence interval.

increase in participant error rates. We, however, found that error rates did not differ between the sound conditions. An important distinction between the current study and Stothart et al. [23] is that the current study employed a generic notification sound, whereas participants in Stothart and colleagues' study received notifications from their personal devices. Though people may likely have a different reaction to their own smartphone notifications, the current study sought to understand the general influence of smartphone notifications on staying focused on the task at hand.

We also found that the oddball effect did not differ between the smartphone notification and control sound trials. Prior work has demonstrated similar null effects of smartphone notifications on behavioral measures of cognitive control [44, 59]. In addition, recent findings demonstrate an habituation effect on reaction times during a cognitive task following the presentation of a phone notification [58]. It could be the case that the behavioral effects of smartphone notifications on cognitive control were lost due to habituation to the auditory stimuli.

To assess for habituation, we examined if null effects were present separately in frequent and rare trials. Frequent trials were presented 768 times whereas rare trials were presented 96 times. Thus, if habituation did occur, a more robust null effect should have been observed for frequent trials. Interestingly, results indicated that smartphone notification sounds were associated with delayed response speed on frequently, but not rarely presented trials. This suggests that habituation to the sound stimuli may not explain the null effect for cognitive control. It could be that greater cognitive demand on rare trials served to prevent distraction from the smartphone notifications. However, during less cognitively demanding frequent trials, smartphone notifications did appear to have an effect on how quickly people responded. This finding is in line with the cognitive load theory of selective attention [14], which posits that when cognitive load is high, such as for rare trials, distractor interference is less likely to occur. Yet, when cognitive load is low, such as for frequent trials, distractor interference is more likely.

**Table 9. Overall P2, N2, and P3 as a function of individual differences in smartphone addiction.**

| ERP | Predictor | b | SE | 95% CI | | df | t | P |
|---|---|---|---|---|---|---|---|---|
| | | | | Lower | Upper | | | |
| P2 at Fz | SAPS[b] | -3.04 | 1.44 | -5.94 | -0.15 | 52 | -2.11 | .039 |
| N2 at F3 | SAPS | -2.47 | 1.37 | -5.22 | 0.28 | 52 | -1.80 | .077 |
| P3 at Pz | SAPS | -2.26 | 1.94 | -6.14 | 1.63 | 52 | -1.16 | .250 |

ERP amplitudes (in microvolts) on the oddball task as a function of individual differences in smartphone addiction. ERPs were time-locked to the presentation of the visual stimulus. [a]N2 is a negative going potential, thus smaller values indicate larger ERP amplitude.

[b]SAPS = Smartphone Addiction Proneness Scale (mean centered). Dependent variable = ERP mean amplitude. CI = Confidence Interval.

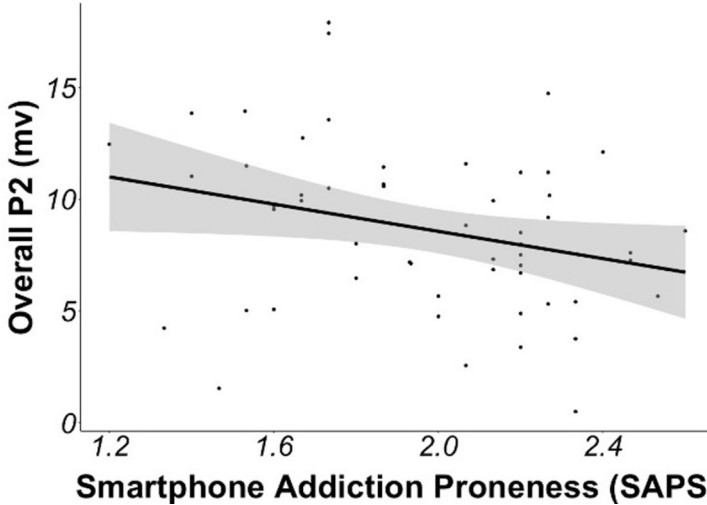

**Fig 7. Correlation between smartphone addiction and P2 ERP.** Correlation between smartphone addiction proneness and the overall P2 mean amplitude (P2 at site Fz within the 150–210 ms time window), indicating that people with higher levels of smartphone addiction show reduced neural activation implicated in early attentional mechanisms.

Contrary to our predictions, we found N2 amplitudes, as well as N2 oddball effect were greater on trials with smartphone compared to control sound trials, suggesting an upregulation of cognitive control when people were exposed to smartphone notifications. This finding is in contrast to prior work which found N2 amplitudes on a cognitive control task to be lowest when a smartphone notification was delivered during the task compared to before or not at all. We did not find this effect of early (P2) and late (P3) attentional processes, which are typically associated with stimulus orienting and categorization. N2 is considered to be involved in cognitive control processes including strategy regulation, immediate action control, novel stimuli detection, and orienting of visual attention [31]. It appears that regardless of the varying degree of cognitive load of trial frequency (i.e., rare vs. frequent), trials with smartphone notification were associated with greater neural activation underlying cognitive control.

## The role of individual differences in smartphone addiction

In this study we found that individual differences in smartphone addiction did not moderate the effects of smartphone notifications as a function of trial type overall. However, examining frequent and rare trials separately, we found that higher levels of smartphone addiction were

**Table 10. Regressions for ERP oddball effect predicted by smartphone addiction.**

| Oddball ERP | Predictor | b | SE | 95% CI Lower | 95% CI Upper | df | t | p |
|---|---|---|---|---|---|---|---|---|
| P2 at Fz | SAPS[b] | 0.78 | 0.72 | -0.65 | 2.22 | 52 | 1.10 | .278 |
| N2 at F3[a] | SAPS | 0.90 | 0.81 | -0.73 | 2.54 | 52 | 1.11 | .273 |
| P3 at Pz | SAPS | -0.51 | 1.36 | -3.24 | 2.22 | 52 | -0.38 | .708 |

P2, N2, and P3 oddball effect as a function of individual differences in smartphone addiction. ERPs were time-locked to the presentation of the visual stimulus.

[a] N2 is a negative going potential, thus smaller values indicate larger ERP amplitude.

[b] SAPS = Smartphone Addiction Proneness Scale (mean centered). Dependent Variable = ERP mean amplitude oddball effect. CI = Confidence Interval.

associated with significantly slower responses on trials with smartphone sounds only on frequent, but not on rare trials, suggesting that for contexts involving greater cognitive load, such as perceptually novel stimuli, smartphone notifications may impact cognitive capacities to a lesser extent.

We also found that higher levels of smartphone addiction were associated with decreased P2 activation overall. P2 activation is considered to reflect a capacity to withdraw attentional resources away from the stimulus [36, 42]. One explanation for this result may be that these individuals are more frequently exposed to notifications on their smartphones, and thus may be less likely to withdraw attention away from the smartphone notifications.

Finally, we found no support for our hypothesis that higher levels of smartphone addiction would be associated with worse cognitive control, reflected by a smaller ERP oddball effect. This suggests that cognitive control assessed as the oddball effect, or the difference in neural activation between frequent and rare trials, did not vary according to levels of reported smartphone addiction. This finding lends no support for contrasting evidence from prior studies demonstrating decreased [44] and increased [43] N2 amplitudes for people higher in smartphone addiction.

Furthermore, our results revealed that higher levels of smartphone addiction were not significantly associated with changes in neural activity between sound conditions. Trials with smartphone notification sounds compared to control sounds did not differ in terms of ERP amplitude as a function of smartphone addiction. Nor were smartphone addiction levels associated with a difference in the oddball effect between sound conditions. Thus, levels of smartphone addiction had no effect on neural activity associated with cognitive control for smartphone notification sounds versus control sounds.

In conclusion, we found partial support to the proposal that cognitive control may be the mechanism for the effect of smartphone notifications reported in the literature, such as their potential unwanted interruption of social interactions [4], walking and driving [5, 6], and educational activities [7, 8]. Further research is needed to clarify the extent to which cognitive control serves as the primary underlying neural mechanism by which these smartphone interruptions negatively impact day-to-day outcomes. It may be that an alternative cognitive process may be more impacted by smartphone notifications, such as working memory, and could thus provide a more complete explanation as to how these interruptions influence people's lives.

From the perspective of cognitive resource allocation, (i.e. cognitive load theory [14]), one explanation of the findings could be that participants developed a mental framework of association between the amount of cognitive resources necessary to attend to smartphone notifications and the amount of top-down control available to sacrifice during task performance. As such, the results suggest that attentional control resources were more easily sacrificed on simple (i.e., frequent) trials, and less so on more difficult (i.e., rare) trials. The lack of a habituation effect, reflected by the null findings for smartphone addiction, further supports this interpretation.

## Limitations and future directions

One limitation of this study was that smartphone addiction was assessed with a self-report measure, although this measure has good psychometric properties [50]. Future work may examine the link between cognitive control and more objective measures of smartphone use, such as data from the smartphone use tracking apps. Furthermore, reported levels of smartphone addiction proneness for the current sample was on the lower end of the possible range. Future work should investigate cognitive effects of smartphone notifications in people who

report higher levels of problematic smartphone use. In addition, the present study included a single unfamiliar control sound. Future studies may add additional control sounds or include trials without sounds for comparison.

Although the N2 is a common approach for measuring electrophysiological markers of conflict monitoring processes, it is not the only ERP component worth examining. The current study did not assess alternative conflict monitoring ERPs, such as the error-related negativity (ERN) component, which is said to capture ERP activity associated with incorrect responses [60, 61]. We were primarily interested in neural activity during correct task responses to measure the neural effect smartphone notifications without the influence of error-related neural activity. Future work, however, needs to examine these alternative ERP components to further elucidate the way smartphone notifications influence top-down control processes.

The current study employed mixed linear model analyses for behavioral measures, while using grand-averaged ERPs due to lack of available data, in turn reducing explainable variance within participants. Future ERP analyses may be improved by including within subject variance into the regression models. Doing so may provide a more complete picture of how smartphone use affects neural processes underlying attention and cognitive control. Along a similar line, many of the effect sizes of the results were small, thus interpretation of these findings should be approached with caution.

Finally, the generalizability of the findings from this study is limited to mostly white undergraduate college students in the Midwest/Southern United States. Though it is critically important to understand the cognitive effects of smartphone use in college students, more diverse samples are needed. It may be that college students in general have relatively greater levels of attention and cognitive control, however those in less cognitively demanding career fields may show different pattern of results. For example, prior work has found a link between smartphone notifications delivered during a cognitive task to be slower for adolescents (vs. mid-life adults) [58]. This suggests that younger populations may be particularly vulnerable to the distractions of smartphone notifications. Future work should investigate how smartphone notifications and smartphone use in general influences the cognitive capacities of participants from different age groups and sociocultural backgrounds. The digital age is characterized by the seemingly constant use of modern technologies, more so now than ever as a result of social distancing and isolation. We must strive to more fully understand how these technologies influence our cognitive functioning. By doing so, we can attempt to maximize the benefits, while minimizing the costs of using these incredibly powerful technologies.

## Author Contributions

**Conceptualization:** Joshua D. Upshaw, Carl E. Stevens, Jr., Giorgio Ganis, Darya L. Zabelina.

**Data curation:** Joshua D. Upshaw.

**Formal analysis:** Joshua D. Upshaw.

**Investigation:** Joshua D. Upshaw.

**Methodology:** Joshua D. Upshaw, Carl E. Stevens, Jr., Giorgio Ganis, Darya L. Zabelina.

**Project administration:** Joshua D. Upshaw.

**Resources:** Giorgio Ganis, Darya L. Zabelina.

**Software:** Carl E. Stevens, Jr.

**Supervision:** Darya L. Zabelina.

**Visualization:** Joshua D. Upshaw.

**Writing – original draft:** Joshua D. Upshaw.

**Writing – review & editing:** Joshua D. Upshaw, Carl E. Stevens, Jr., Darya L. Zabelina.

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
