## [Decision Letter · Decision Letter 0]

14 Jul 2022

PONE-D-22-03238The hidden cost of a smartphone: The effects of smartphone notifications on cognitive control from a behavioral and electrophysiological perspective.PLOS ONE

Dear Dr. Upshaw,

Thank you for submitting your manuscript to PLOS ONE; I sincerely apologise for the unusually delayed review timeframe. Your manuscript has been assessed by one reviewer, whose comments are appended below. After careful consideration, we feel that it has merit but does not fully meet PLOS ONE’s publication criteria as it currently stands. Although the reviewer comments that "The use of N2/P2/P3 components to index cognitive control changes in response to smartphone notification was interesting and and original" and "The statistical analysis were careful and accurate", they raise a number of concerns regarding the description of the rationale and methodology, as well as the discussion of the results. Therefore, we invite you to submit a revised version of the manuscript that addresses the points raised during the review process. Please note that we have only been able to secure a single reviewer to assess your manuscript. We are issuing a decision on your manuscript at this point to prevent further delays in the evaluation of your manuscript. Please be aware that the editor who handles your revised manuscript might find it necessary to invite additional reviewers to assess this work once the revised manuscript is submitted. However, we will aim to proceed on the basis of this single review if possible.

We look forward to receiving your revised manuscript.

Kind regards,

Emily Chenette

Editor in Chief

PLOS ONE

Reviewers' comments:

Reviewer's Responses to Questions

**Comments to the Author**

1. Is the manuscript technically sound, and do the data support the conclusions?

Reviewer #1: Yes

2. Has the statistical analysis been performed appropriately and rigorously? 

Reviewer #1: Yes

3. Have the authors made all data underlying the findings in their manuscript fully available?

Reviewer #1: No

4. Is the manuscript presented in an intelligible fashion and written in standard English?

Reviewer #1: Yes

5. Review Comments to the Author

Reviewer #1: The paper adds to the increasing number of publications on the cognitive effects of smart phone use. Specifically, the author investigate its putative effects on attention and cognitive control. The authors use a clever design coupling a global/local task with an oddball paradigm to examine whether trials preceded by a smartphone notification sounds would be subjected to greater involuntary distraction than trials preceded by control sounds. Changes in ERP components (N2, P2, P3) that underlie attentional orienting and cognitive control as well as changes in behavioral responses (accuracy and response times) were collected and carefully analyzed through multilevel linear models. In addition, a smart phone addiction questionnaire was collected and included in the analysis.

The results suggest that smartphone notification do not impair cognitive control processes. Rather, they are associated with deploying less control only on the easiest conditions, while control on the hardest trials seems unimpaired, as shown by both behavioral data and P2 and P3 findings. This result seems to be confirmed by the lack of reliable interactions between scores on the smartphone addiction scale and cognitive control effects (behavioral or ERPs). Even the few significant effects (effects of notification on common targets, effects on N2 component) seem small in term of effect sizes.

Overall, I found the paper interesting and well-written. The use of N2/P2/P3 components to index cognitive control changes in response to smartphone notification was interesting and and original. The statistical analysis were careful and accurate. However, I do have some concerns, which I outline below.

1. The paper does not seem to want to speculate about what these results mean. A tentative explanation would be that participants have learned an association between smartphone sounds and the amount of attentional control they can remove from the task, so that easier task condition can be sacrificed (to pay attention to the smartphone) but difficult trials cannot. The lack of habituation effects suggests that such an association is also well ingrained.

2. The justification for the neural (ERP) effects is somewhat thin. The various ERPs seemed to be introduced as a series of “this seems relevant, and this seems relevant also…”. But why were they chosen? Other ERPs could be chosen and justified. For instance, one could argue that ERNs and FRNs, which are associated to on-line adjustments of control, could have been equally relevant. Thus, the selection must be based on some assumptions of what facet of cognitive control the authors intend to focus on. I am imagining a section, in the introduction, that sounds more or less like “we considered three ERPs that are commonly accepted in the literature as being related to attention. Each of them [or all of them] tackles one specific aspect (...). Other ERPs were not included because (...)”.

3. A more general question is the nature of cognitive control that is examined in this paper. Like "Attention", "Cognitive control" is a bit of a slippery term, and different authors imply different things. It would be helpful if the authors could situate a bit better their ideas about control in the larger control/execute function literature, which would also help justify the choice of the Global/Local task (as opposed to, for example, the Flanker task, the Stroop task, the CPT task, all of which could have been mixed with the oddball paradigm).

4. Speaking of which, the experimental paradigm is described insufficiently well. Without the figure, it is impossible to understand whether the composing letters are all the same or not, and which letters were used. Even with the figure, it is not explicitly what was the exact task (to identify the letter among the small, the big, or both?), although the reader can make some reasonable assumptions.

5. The DOI link to the data (https://doi.org/10.7910/DVN/0NIJWN) does not seem to exist or to be linked to any online resource.. Thus, data cannot be accessed

Minor points:

6. “The control sound was created in Audacity (v. 2.2.2, [41]), and was a square wave tone closely matched to the smartphone sound in duration, volume, and sound similarity.” (page 15). Is it possible to include a figure of the spectral densities or waveforms of the three sounds used in the experiment?

7. I cannot understand the sentence “or not at all 10% of the time (non-targets; Fig 1B).”??? [Page 14]

6. PLOS authors have the option to publish the peer review history of their article (what does this mean?). If published, this will include your full peer review and any attached files.

Reviewer #1: No

---

## [Author Response · Author response to Decision Letter 0]

25 Aug 2022

Dr. Emily Chenette August 25, 2022

PLOS ONE

Dear Dr. Chenette,

Thank you for your consideration of our manuscript entitled “The hidden cost of a smartphone: The effects of smartphone notifications on cognitive control from a behavioral and electrophysiological perspective.” for publication consideration in the PLOS ONE. We were very pleased with the reviews we’ve and made changes according to yours and the reviewers suggestions (in blue font in the manuscript). We detail these changes below. 

Response to Editor and Journal Requirements .

Author Response:

We have examined our manuscript and made several adjustments and to the best of our knowledge the revised submission meets PLOS ONE’s style requirements and file naming conventions. 

Author Response:

We have developed a private github repository that will store the data to be shared and will be made public upon publication acceptance. This is the link to the github repository: https://github.com/jupshaw/SMARTPHONES-AND-COGNITIVE-CONTROL

Author Response:

We have included an Ethics statement subsection in the methods section. 

“Ethics statement

The study was approved by the local Institutional Review Board at the University of Arkansas and was assigned the protocol number 1807134340. All participants provided written informed consent to participate. Participants were compensated with course credit.” (pp. 9)

Reviewer 1 Comments to the Author

Reviewer #1: The paper adds to the increasing number of publications on the cognitive effects of smart phone use. Specifically, the author investigate its putative effects on attention and cognitive control. The authors use a clever design coupling a global/local task with an oddball paradigm to examine whether trials preceded by a smartphone notification sounds would be subjected to greater involuntary distraction than trials preceded by control sounds. Changes in ERP components (N2, P2, P3) that underlie attentional orienting and cognitive control as well as changes in behavioral responses (accuracy and response times) were collected and carefully analyzed through multilevel linear models. In addition, a smart phone addiction questionnaire was collected and included in the analysis.

The results suggest that smartphone notification do not impair cognitive control processes. Rather, they are associated with deploying less control only on the easiest conditions, while control on the hardest trials seems unimpaired, as shown by both behavioral data and P2 and P3 findings. This result seems to be confirmed by the lack of reliable interactions between scores on the smartphone addiction scale and cognitive control effects (behavioral or ERPs). Even the few significant effects (effects of notification on common targets, effects on N2 component) seem small in term of effect sizes.

Overall, I found the paper interesting and well-written. The use of N2/P2/P3 components to index cognitive control changes in response to smartphone notification was interesting and and original. The statistical analysis were careful and accurate. However, I do have some concerns, which I outline below.

1. The paper does not seem to want to speculate about what these results mean. A tentative explanation would be that participants have learned an association between smartphone sounds and the amount of attentional control they can remove from the task, so that the easier task trials can be sacrificed (to pay attention to the smartphone) but difficult trials cannot. The lack of habituation effects suggests that such an association is also well ingrained.

Author response:

We appreciate the reviewer comments on the need for a stronger speculation about the meaning of our results. We added a new paragraph to the conclusion section to address this concern. 

“From the perspective of cognitive resource allocation, (i.e., cognitive load theory; Lavie et al., 2014) one explanation of the findings could be that participants developed a mental framework of association between the amount of cognitive resources necessary to attend to smartphone notifications and the amount of top-down control available to sacrifice during task performance. As such, the results suggest that attentional control resources were more easily sacrificed on simple (i.e., frequent) trials, and less so on more difficult (i.e., rare) trials. The lack of a habituation effect, reflected by the null findings for smartphone addiction, further supports this interpretation.” (pp. 30)

2. The justification for the neural (ERP) effects is somewhat thin. The various ERPs seemed to be introduced as a series of “this seems relevant, and this seems relevant also…”. But why were they chosen? Other ERPs could be chosen and justified. For instance, one could argue that ERNs and FRNs, which are associated to on-line adjustments of control, could have been equally relevant. Thus, the selection must be based on some assumptions of what facet of cognitive control the authors intend to focus on. I am imagining a section, in the introduction, that sounds more or less like “we considered three ERPs that are commonly accepted in the literature as being related to attention. Each of them [or all of them] tackles one specific aspect (...). Other ERPs were not included because (...)”.

Author response:

We appreciate the reviewer bringing this to our attention and we agree that further discussion for the justification of using the specific ERPs is warranted. We included the following pieces of text in the Introduction and Discussion sections, under “The Present Study” and “Limitations and future directions” subheadings, respectively, to address this issue.

“Cognitive control was measured using the oddball effect, which is calculated by subtracting reaction times (RT) and ERP amplitudes on frequent target trials from rare target trials. Better cognitive control is considered to be reflected by a smaller RT oddball effect and a larger ERP oddball effect [32]. We examined three ERPs, the P200 (P2), N200 (N2), and P300 (P3), which are commonly accepted in the literature as underlying neural markers of electrical cortical activation associated with attention and cognitive control processes [30,33,34]. 

This paradigm and various adaptations have been used in prior studies indicating a family of frontocentral N2 components related to cognitive control [33]. The N2 ERP component is the second negative peak along the average ERP waveform which generally occurs between 200-350 ms after stimulus onset near frontocentral and central electrode site. N2 is considered to be involved in strategy regulation, feedback processing, immediate action control, novel stimuli detection, and visual attention orienting [33]. Though multiple N2 subcomponents exist, the current study focused on a frontocentral N2 component related to cognitive control processes of response inhibition, response conflict, and error monitoring. This anterior N2 component is said to be generated from the anterior cingulate gyrus [35] and is associated with top-down control of attention [36]. Based on previous literature, we expected that participants would respond more slowly and would show a smaller N2 oddball effect (i.e., worse cognitive control) on trials with the smartphone notification (vs. control) sounds.

We examined the P2 ERP as it is likely to be affected by exposure to smartphone notifications. The P2 ERP is the second positive peak along the average ERP waveform which generally occurs between 150-250 ms after stimulus onset near frontal electrode sites [34]. P2 is considered to reflect stimulus monitoring and early attention classification processes and has been shown to demonstrate differential activation between target stimuli conditions in oddball paradigms assessing capacities to withdraw attentional resources away from stimuli [37],[38]. P2 is said to be generated largely as a result from activation within the reticular activating system [39] as a response to input from sensory modalities [34]. If it is the case that smartphone notifications “capture” people’s attention, trials with smartphone notification (vs. control) sounds should elicit a larger P2 ERP.

Among oddball paradigms, an anterior N2 component to frequent targets is often observed in combination with a posterior P3 component to distractor targets, suggesting cognitive processes involved in contextual and memory updating (Debener et al., 2005; Spencer et al., 2001). The P3 ERP component is the third positive peak along the average ERP waveform which generally occurs between 250-500 ms after stimulus onset near large frontoparietal scalp electrode networks [30]. P3 can be divided into two subcomponents. P3 is said to originate from frontal lobe activation for attention-driven stimuli processing, particularly for task-irrelevant neural activity elicited during target stimulus processing [30]. P3 has been considered to be a late cognitive component involved in endogenous decision-making and stimulus categorization originating from the dorsolateral prefrontal cortex (DLPFC) which communicates with the cingulate cortex and parietal structures [35]. P2 and N2 reflect early cognitive processes and are likely to be influenced by exogenous smartphone notifications. P3, on the other hand, should not be affected by smartphone notifications, as P3 is thought to reflect late cognitive processes involved in endogenous decision-making and stimulus categorization [32].” (pp. 6-8)

“Although the N2 is a common approach for measuring electrophysiological markers of conflict monitoring processes, it is not the only ERP component worth examining. The current study did not assess alternative conflict monitoring ERPs, such as the error-related negativity (ERN) component, which is said to capture ERP activity associated with incorrect responses [60,61]. We were primarily interested in neural activity during correct task responses to measure the neural effect smartphone notifications without the influence of error-related neural activity. Future work, however, needs to examine these alternative ERP components to further elucidate the way smartphone notifications influence top-down control processes.” (pp. 31)

3. A more general question is the nature of cognitive control that is examined in this paper. Like "Attention", "Cognitive control" is a bit of a slippery term, and different authors imply different things. It would be helpful if the authors could situate a bit better their ideas about control in the larger control/execute function literature, which would also help justify the choice of the Global/Local task (as opposed to, for example, the Flanker task, the Stroop task, the CPT task, all of which could have been mixed with the oddball paradigm).

Author response:

We now include the following information in the introduction of the manuscript to further situate our ideas about cognitive control:

“The present study aimed to examine the extent to which smartphone notifications influence cognitive control and attention on an adapted Local/Global hierarchical letter three-stimulus oddball paradigm using event-related potentials (ERPs), and behavioral performance. The Local/Global task requires an individual to attentionally reorient to and update working memory to accurately respond to the presence of a target letter while monitoring for the presence of rare distractor letters presented at opposite levels of visual attention [29,30]).The rare target letter presents an exogenous salient singleton requiring increased recruitment of attentional and cognitive control resources, specifically those necessary for conflict monitoring [31]. Cognitive control paradigms such as the Stroop or Erickson Flanker task also measure conflict monitoring, however, the paradigm in the current study was chosen to heighten engagement of early attentional orienting mechanisms. While monitoring for conflict between frequent, rare, and non-target trials, participants were required to ignore inconsistent visual information between hierarchically nested visual stimuli presented at opposing levels of local or global attention. Therefore, this paradigm allowed us to simultaneously measure the effects of smartphone notifications on attention and cognitive control.” (pp. 5-6)

4. Speaking of which, the experimental paradigm is described insufficiently well. Without the figure, it is impossible to understand whether the composing letters are all the same or not, and which letters were used. Even with the figure, it is not explicitly what was the exact task (to identify the letter among the small, the big, or both?), although the reader can make some reasonable assumptions.

Author response:

We now present a more clear description of the task to address this issue:

“Participants were asked to indicate the presence of a target letter by pressing either Yes (1 key) for “Target letter is present,” or No (2 key) for “Target letter is not present” using their right hand on a standard keyboard digit pad. Participants were instructed to detect the presence of lack of presence of the target letter regardless of the size of the letter. Participants completed two practice blocks consisting of 9 trials each. Visual feedback was provided for response accuracy on practice trials (“Correct” or “Incorrect”). Following practice trials, participants completed 16 experimental blocks (960 trials). Before each block, a “target letter” screen was displayed on the computer monitor for 12 seconds. The “target letter” screen used a single red letter (twice in size as the local letters) to identify the specific target letter participants would be aiming to detect in the following block of trials. 

Each block of the task consisted of 60 trials presented in pseudo-random order, ensuring that an equal number of sound stimuli were presented on frequent, rare, and non-target trials. On a given block, target letters were displayed on the screen at either the local or global attentional level on 80% of trials, referred to as frequent trials. During the same task block, target letters of the opposite level of attention were displayed on 10% of trials, referred to as rare trials. The final 10% of trials did not include a target letter, referred to as non-target trials; Fig 1B). Global and local letter stimuli were counterbalanced such that big “H”s, “E”s, “S”s, and “A”s, were composed of uniform (never mixed within a single letter) sets of smaller letters for an equal presentation of hierarchical letter combinations across the 4 four possible letters to choose from. For example, on a given trial, a big “H” would be comprised of all small “S”s, but never small “S”s and small “E”s. Each block of 60 individual trials was followed by a self-paced break period.” (pp. 10-11)

5. The DOI link to the data (https://doi.org/10.7910/DVN/0NIJWN) does not seem to exist or to be linked to any online resource. Thus, data cannot be accessed

Author response: 

We created a publicly available github repository for the project where the data and other coding script materials will be made available upon acceptance of this manuscript for publication.

https://github.com/jupshaw/SMARTPHONES-AND-COGNITIVE-CONTROL

Minor points:

6. “The control sound was created in Audacity (v. 2.2.2, [41]), and was a square wave tone closely matched to the smartphone sound in duration, volume, and sound similarity.” (page 15). Is it possible to include a figure of the spectral densities or waveforms of the three sounds used in the experiment?

Author response:

We added the text below and created the following figure to illustrate the spectral densities and waveforms of the auditory stimuli.

“Consequent examination of the lawnmower sound acoustic waveform spectrum revealed unintended technical confounds (e.g. stereoscopic inconsistency creating a perception of spatial movement (Fig 2C) and was not used in subsequent analyses. Sound stimuli were presented pseudo-randomly.” (pp. 12)

“Fig 2. Frequency spectral densities and signal waveforms of the auditory stimuli.” (pp. 12)

7. I cannot understand the sentence “or not at all 10% of the time (non-targets; Fig 1B).”??? [Page 14]

Author response:

We edited this sentence to improve its readability and comprehension.

“On a given block, target letters were displayed on the screen at either the local or global attentional level on 80% of trials, referred to as frequent trials. During the same task block, target letters of the opposite level of attention were displayed on 10% of trials, referred to as rare trials. The final 10% of trials did not include a target letter, referred to as non-target trials; Fig 1B).” (pp. 10-11)

---

## [Decision Letter · Decision Letter 1]

10 Oct 2022

PONE-D-22-03238R1The hidden cost of a smartphone: The effects of smartphone notifications on cognitive control from a behavioral and electrophysiological perspective.PLOS ONE

Dear Dr. Upshaw,

Thank you for submitting your manuscript to PLOS ONE. After careful consideration, we feel that it has merit but does not fully meet PLOS ONE’s publication criteria as it currently stands. Therefore, we invite you to submit a revised version of the manuscript that addresses all the points raised by the reviewer including the data access.

We look forward to receiving your revised manuscript.

Kind regards,

Francesco Di Russo, Ph.D.

Academic Editor

PLOS ONE

Journal Requirements:

Reviewers' comments:

Reviewer's Responses to Questions

**Comments to the Author**

1. If the authors have adequately addressed your comments raised in a previous round of review and you feel that this manuscript is now acceptable for publication, you may indicate that here to bypass the “Comments to the Author” section, enter your conflict of interest statement in the “Confidential to Editor” section, and submit your "Accept" recommendation.

Reviewer #1: All comments have been addressed

2. Is the manuscript technically sound, and do the data support the conclusions?

Reviewer #1: Yes

3. Has the statistical analysis been performed appropriately and rigorously? 

Reviewer #1: Yes

4. Have the authors made all data underlying the findings in their manuscript fully available?

Reviewer #1: No

5. Is the manuscript presented in an intelligible fashion and written in standard English?

Reviewer #1: Yes

6. Review Comments to the Author

Reviewer #1: The authors have responded to all of my previous comments to my satisfaction. My only concern is that the Github repo is still private, so I had not had a chance to fully examine the LMER model and make sure I understand them correctly. Per PLOSONE policy, the repository MUST be made public before publication (I would also recommend sharing it through OSF).

Besides that, I have only four VERY MINOR comments that would be great to see fixed before publication:

1. “An intraclass correlation of r = 0.14”. I noted this in my previous review, I am not sure if the authors actually mean ICC (suchas ICC = 0.14) or Pearson correlation.

2, The participant info (number, age, proportion female) is repeated twice, in the special Participant subsection and then again below.

3. Page 15 “ This model was found to provide the best fit of the data.” Best compared to… What? Maybe other, simpler LMER models?

4. Page 16. After explaining the logistic regression model for accuracy, the authors note that “ Error rates were assessed using the same predictor variables .”. But… Aren’t error rates just the complement of accuracies? And doesn’t the logistic model already provide a complete account?

7. PLOS authors have the option to publish the peer review history of their article (what does this mean?). If published, this will include your full peer review and any attached files.

Reviewer #1: No

---

## [Author Response · Author response to Decision Letter 1]

18 Oct 2022

Editor and Journal Requirements .

Author Response:

We have double checked all articles to ensure that they have not been retracted. We employed Zotero reference manager and google scholar for retraction checks.

Reviewer Comments to the Author

Reviewer #1: The authors have responded to all of my previous comments to my satisfaction. My only concern is that the Github repo is still private, so I had not had a chance to fully examine the LMER model and make sure I understand them correctly. Per PLOSONE policy, the repository MUST be made public before publication (I would also recommend sharing it through OSF).

Author Response:

We have created a publicly available OSF repository that includes the R scripts and data files to improve transparency. This can be accessed here: https://osf.io/bj7zf/

The data availability statement will also be adjusted accordingly.

MINOR comments:

1. “An intraclass correlation of r = 0.14”. I noted this in my previous review, I am not sure if the authors actually mean ICC (such as ICC = 0.14) or Pearson correlation.

Author Response:

Thank you for catching this typo. We have adjusted the sentence appropriately.

“An intraclass correlation (ICC) of 0.14 for RT within participant was found, warranting the use of mixed linear models.” (p. 15)

2. The participant info (number, age, proportion female) is repeated twice, in the special Participant subsection and then again below.

Author Response:

We have removed the unnecessary participant info as it did not provide additional useful information.

“Four participants were excluded from behavioral data analyses due to technical issues or poor performance on the oddball task (i.e., errors or RTs exceeding +/- 2.5 SD). The final sample for behavioral analyses included 69 participants. ERP data for 19 participants were excluded because of technical issues, or for having uncorrectable artifacts greater than 25% of total trials [48]. The final sample for ERP analyses included 54 participants.” (p. 12)

3. Page 15 “ This model was found to provide the best fit of the data.” Best compared to… What? Maybe other, simpler LMER models?

Author Response:

Thank you for pointing out this area of confusion. We have adjusted the analytical strategy section to improve the clarity on these points.

“For RTs, analyses were conducted using linear mixed effect regression (LMER) models with random slopes for the trial frequency condition (rare and frequent) and random intercepts for each participant to account for within-subject variance in RT across all trials (lmerTest v. 3.1.1) [57]. This model was found to provide the best fit of the data relative to simpler LMER models.” (p. 15)

4. Page 16. After explaining the logistic regression model for accuracy, the authors note that “ Error rates were assessed using the same predictor variables .”. But… Aren’t error rates just the complement of accuracies? And doesn’t the logistic model already provide a complete account?

Author Response: 

Thank you for pointing out this redundant information. We have adjusted the paragraph to improve clarity.

“For the accuracy analyses, we conducted generalized linear mixed effects regression model with error rate as the dependent variable, random slopes for trial frequency, and random intercepts for participant. We used a bound optimization by quadratic approximation with a binomial family distribution of 0 for correct and 1 for incorrect trials.” (p. 16)

---

## [Editor Report · Decision Letter 2]

24 Oct 2022

The hidden cost of a smartphone: The effects of smartphone notifications on cognitive control from a behavioral and electrophysiological perspective.

PONE-D-22-03238R2

Dear Dr. Upshaw,

We’re pleased to inform you that your manuscript has been judged scientifically suitable for publication and will be formally accepted for publication once it meets all outstanding technical requirements.

Kind regards,

Francesco Di Russo, Ph.D.

Academic Editor

PLOS ONE
---

## [Editor Report · Acceptance letter]

9 Nov 2022

PONE-D-22-03238R2 

The hidden cost of a smartphone: The effects of smartphone notifications on cognitive control from a behavioral and electrophysiological perspective. 

Dear Dr. Upshaw:

I'm pleased to inform you that your manuscript has been deemed suitable for publication in PLOS ONE. Congratulations! Your manuscript is now with our production department. 

Kind regards, 

on behalf of

Prof. Francesco Di Russo 

Academic Editor

PLOS ONE